# Exploring the Impact of Model Scaling on Parameter-efficient Tuning

**Yusheng Su**[1*]**, Chi-Min Chan**[1*]**, Jiali Cheng**[2]**, Yujia Qin**[1]**, Yankai Lin**[3]**, Shengding Hu**[1]**,
**Zonghan Yang**[1]**, Ning Ding**[1]**, Xingzhi Sun**[4]**, Guotong Xie**[4]**, Zhiyuan Liu**[1†]**, Maosong Sun**[1+]

[1]Department of Computer Science and Technology, Tsinghua University
[2]University of Massachusetts Lowell
[3]Gaoling School of Artificial Intelligence, Renmin University
[4]Ping An Technology
yushengsu.thu@gmail.com

## Abstract

Parameter-efficient tuning (PET) methods can
effectively drive extremely large pre-trained
language models (PLMs) by training only min-
imal parameters. Different PET methods uti-
lize different manually designed tunable mod-
ules. In small PLMs, there are usually notice-
able performance differences among PET meth-
ods. Nevertheless, as the model scale increases,
the performance differences become marginal.
Hence, we hypothesize that model scaling miti-
gates the impact of design differences on PET
methods. To investigate this hypothesis, we
introduce a more flexible PET method called
Arbitrary PET (APET) method. The APET
method is compatible with a tunable module,
which consists of any number of parameters dis-
tributed in arbitrary positions. Then, we utilize
it and conduct experiments on 11 NLP tasks
across 3 representative PLMs. Our investiga-
tions reveal that model scaling (1) mitigates
the effects of the positions of tunable param-
eters on performance, and (2) enables tuning
methods to achieve performance comparable to
full-parameter fine-tuning by optimizing fewer
tunable parameters. Intriguingly, we also ob-
serve that tuning methods optimize the similar
number of tunable parameters to exceed ran-
dom guess performance on different tasks. We
collectively discuss this phenomenon and the
two aforementioned findings from an optimiza-
tion perspective to understand the underlying
mechanisms. These conclusions enhance our
understanding of the impact of model scaling
on PET and assist in designing more effec-
tive and efficient PET methods for PLMs of
different scales. The source code can be ob-
tained from this GitHub repository: https://
github.com/yushengsu-thu/PET_Scaling.

## 1 Introduction

Pre-trained language models (PLMs), such as GPT
(Radford et al., 2018), BERT (Devlin et al., 2019),

---

* The first two authors contributed equally.
† Corresponding author: Z.Liu and M.Sun.

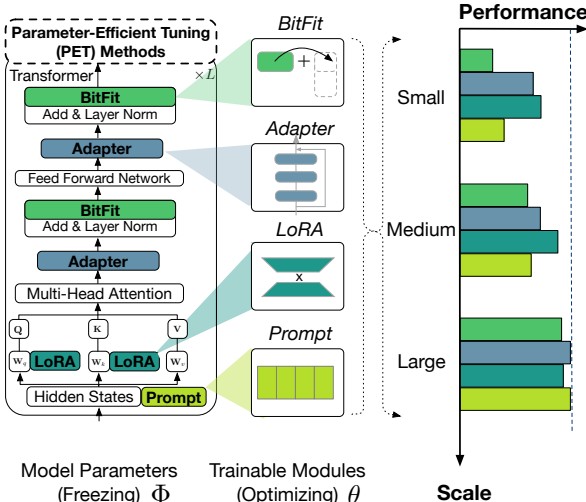

Figure 1: Different PET methods have distinct tunable
modules, which typically result in noticeable perfor-
mance differences. However, as the model scale in-
creases, these differences become less significant.

and T5 (Raffel et al., 2020), have achieved great
success on various natural language processing
(NLP) tasks. Despite their effectiveness, fine-
tuning (FT) these large-scale PLMs with full param-
eters incurs both unaffordable computational and
storage costs. To solve this problem, researchers
have proposed a series of parameter-efficient tun-
ing (PET) methods (Houlsby et al., 2019a; Li and
Liang, 2021; Mahabadi et al., 2021a; Lester et al.,
2021; Mahabadi et al., 2021b; Hu et al., 2022a;
Ben Zaken et al., 2022; He et al., 2022b) which
only update an assigned tunable module consist-
ing of minimal parameters while freezing the rest
parameters in a PLM during model adaptation.

Although these existing representative PET
methods can reduce computational and storage
costs, there are usually noticeable performance dif-
ferences among these representative PET methods
on downstream tasks. Intriguingly, as the scale
of a PLM increases, the performance differences
among PET methods become narrower, as illus-

trated in Figure 1. These findings are interesting and worth exploring because the existing representative PET methods are designed with disparate philosophies, e.g., tunable modules that are composed of **different numbers** of tunable parameters distributed in **arbitrary positions**. Hence, we hypothesize that *model scaling mitigates the effects of the above design differences among the PET methods on performance*. To validate this hypothesis, we further conduct two lines of ablation analyses:

(A1) Whether the model scale mitigates the performance differences resulting from the position of tunable parameters.

(A2) Whether the model scale mitigates the performance differences resulting from the number of tunable parameters.

However, solely investigating the four representative PET methods (see Figure 1) might be insufficient to encompass an adequate range of parameter positions for the ablation analyses (A1). Additionally, the tunable modules of these four PET methods are constrained to be composed of layer-level tensors or matrices, making it challenging to precisely control the number of tunable parameters at the fine-grained (parameter) level in the ablation analyses (A2). To facilitate the ablation analyses, we develop a more flexible **A**rbitrary **P**arameter-**E**fficient **T**uning (APET) method (§ 5.1), which can be compatible with any number of tunable parameters distributed in arbitrary positions.

In analysis (A1), we compare the performance of APET methods with an equal number of tunable parameters distributed in different positions. Based on the experimental results, we observe smaller differences in the performance of these APET methods on larger models. This finding suggests that scaling the model *mitigates the effects caused by the position of tunable parameters on performance*.

In analysis (A2), we compare the performance of the same APET methods with varying numbers of tunable parameters. Based on the experimental results, we observe that model scaling does not mitigate the effects caused by the number of tunable parameters on performance. Furthermore, we have observed two interesting phenomena when the number of tunable parameters reaches two thresholds: the high threshold and the low threshold. When the number of tunable parameters equals the high threshold, APET methods can achieve the full-parameter fine-tuning performance of the corresponding backbone model, and the high threshold

tends to be lower on the larger models. Namely, *PET methods can optimize fewer tunable parameters to achieve full-parameter fine-tuning performance on the larger models*. On the other hand, when the number of tunable parameters exceeds the low parameter threshold, all APET methods outperform random guess performance. We find that the low thresholds are nearly identical across the same models, even for different tasks. This suggests that *across different tasks, PET methods can optimize a similar number of tunable parameters on the same PLM to surpass random guess performance*.

In summary, we introduce a more flexible PET methods - APET methods - to conduct the extensive ablation analyses and reveal the impact of model scaling on PET design, e.g., (1) the position of tunable parameters (§ 5.2) and (2) the number of tunable parameters (§ 5.3). (3) Furthermore, we discuss the findings of ablation analyses from the perspective of optimization (§ 6). We hope these conclusions not only encourage more researchers to explore the impact of model scaling on tuning methods from a theoretical perspective, but also provide guidance for designing tuning methods for models of different scales.

## 2 Related Work

**Parameter-Efficient Tuning (PET) Methods** With larger PLMs continuously being developed, fine-tuning all of the parameters and storing the adapted weights become increasingly cumbersome. To address the issue, researchers propose PET methods which keep most of the parameters of PLMs frozen and optimize only a tunable module consisting of a few parameters during downstream adaptation. Over the recent years, many different designs of PET methods have emerged. For instance, some PET methods insert the external tunable modules after the feed-forward and attention layers in a PLM (Houlsby et al., 2019a; Pfeiffer et al., 2021; Mahabadi et al., 2021c); others prepend the tunable modules into attention layers (Li and Liang, 2021; Hu et al., 2022a) or the embedding layer (Lester et al., 2021). Another line of PET method selects the existing parameters in a PLM (Ben Zaken et al., 2022; Guo et al., 2021) as the tunable module to optimize. To further enhance the performance of PET methods, some works propose automatic selection strategies (Hu et al., 2022c; Chen et al., 2023; Lawton et al., 2023; Zhou et al., 2023) for tunable parameters.

| PET Methods | Unified View of PET Methods | Positions of Tunable Modules $\theta = \{\mathbf{W}_1, \mathbf{W}_2, ..., \mathbf{W}_p\}$ |
|---|---|---|
| Prompt (Lester et al., 2021) Adapter (Houlsby et al., 2019a) LoRA (Hu et al., 2022a) BitFit (Ben Zaken et al., 2022) | $\mathbf{h}^{out} = f(\mathbf{h}^{in}) + \Delta\mathbf{h}$ | **W** will be **concatenated** to **input hidden states** **W** will be **plugged** between **SelfAttn./FFN. layers** **W** will be **plugged** into **SelfAttn layers** **W** will be **add** into **Bias terms** |

Table 1: We uniformly re-frame the transformations of PET methods as modifications $\Delta\mathbf{h}$ of specific hidden states in the corresponding PLM layer ($f$) where **W** is introduced in computing $\Delta\mathbf{h}$, as suggested by He et al. (2022a); Hu et al. (2022c). Each PET method has $p$ tunable weights **W** in designed positions. Hence, we represent each PET tunable module as $\theta = \{\mathbf{W}_1, \mathbf{W}_2, ..., \mathbf{W}_p\}$.

Although these PET methods have distinct tunable modules, they can be unified into a similar form. He et al. (2022a) formalize PET methods as a unified framework to study the connections among PET methods. Yi et al. (2022) also conduct the same study and further indicate that the optimization of different PET methods can be unified in a similar subspace. In this paper, we leverage these unified perspectives to explain the impact of model scaling on PET in the final discussion (§ 6).

**The Power of Model Scaling** With the scaling of model size, PLMs emerge numerous capabilities, including reasoning ability (Wei et al., 2022b,a), and can achieve state-of-the-art results in various understanding and generation tasks (Du et al., 2022; Chowdhery et al., 2022).

In the adaption perspective, some researchers find that performing some PET methods (Lester et al., 2021; Ding et al., 2023; Su et al., 2022) on large-scale models can almost achieve the full-parameter fine-tuning performance. In this paper, we further find that as the model scale increases, the performance differences among distinct PET methods become smaller (§ 4). Hence, we study the impact of model scaling on PET methods (§ 5) to fathom this phenomenon and explain it from the optimization perspective (§ 6).

## 3 Preliminary

In this section, we first introduce the Transformer framework (§ 3.1) and the most representative PET (§ 3.2).

### 3.1 Transformer Framework

The Transformer model (Vaswani et al., 2017) is the mainstream architecture for most powerful PLMs. The model is stacked of $L$ blocks, each of which consists of a sequence of layers, including self-attention and feed-forward network. During the forward pass through each block, the input hidden

state is applied with the sequence of layers. For simplicity, we formalize the transformation of each layer as

$$\mathbf{h}^{out} = f(\mathbf{h}^{in}). \tag{1}$$

Under the layer as the operator $f$, the input hidden state $\mathbf{h}^{in} \in \mathbb{R}^{s \times d_{in}}$ is transformed into the output hidden state $\mathbf{h}^{out} \in \mathbb{R}^{s \times d_{out}}$, where $s$ is the input length and $d_{in}, d_{out}$ are dimensions.

### 3.2 Parameter Efficient Tuning (PET)

Different PET methods[1] are equipped with diverse modules $\theta$ as shown in Figure 1. These modules are composed of tunable parameters **W** that modify the original layers and the corresponding transformations in PLMs. To make comparisons, we follow the unified view (He et al., 2022a; Hu et al., 2022c) to re-frame the transformations of all PET methods as the modifications $\Delta\mathbf{h}$ of specific hidden states in the corresponding PLM's layers as follows:

$$\mathbf{h}^{out} = f(\mathbf{h}^{in}) + \Delta\mathbf{h}. \tag{2}$$

In the training process, given a downstream task $\mathcal{D} = \{X, Y\}$, we only optimize all tunable parameters of the module $\theta$ for each PET method to generate desired outputs $Y$ of a downstream task while freezing the rest of the parameters $\Phi$ in a PLM $\mathcal{M}$, as shown in Figure 1[2]. Formally, the training objective is to minimize $\mathcal{L}$ as follows:

$$\min_\theta \mathcal{L}(\mathcal{M}_{(\Phi,\theta)}(X), Y). \tag{3}$$

## 4 Main Experiments

To explore the impact of model scaling on these PET methods, we first introduce the investigated tasks, PLMs, and settings of the existing representative PET methods in the experiments (§ 4.1), and then report the main experimental results (§ 4.2).

---

[1]More implementation details are left in appendix B.
[2]The manipulations, including addition, concatenation, and plugging, are discussed in § 5.1.

## 4.1 Experimental Settings

**Investigated NLP Tasks**  We investigate 11 tasks, which can be divided into 5 categories: (1) *Sentiment Analysis* (SA), including SST-2 (Socher et al., 2013), IMDB (Maas et al., 2011), and Rotten Tomatoes (Pang and Lee, 2005); (2) *Natural Language Inference* (NLI), including MNLI (Williams et al., 2018), QNLI (Wang et al., 2019), and RTE (Bos and Markert, 2005); (3) *Paraphrase Identification* (PI), including MRPC (Dolan and Brockett, 2005) and QQP (Sharma et al., 2019); (4) *Question Answering* (QA), including NQ-Open (Lee et al., 2019); (5) *Summarization* (SUM), including SAMSum (Gliwa et al., 2019) and Multi-News (Fabbri et al., 2019). More details are in appendix A.

**Investigated PLMs**  We will experiment on three series of PLM backbones: BERT (Devlin et al., 2019), BLOOM (Scao et al., 2023), and T5 (Raffel et al., 2020) representing encoder-based model, decoder-based model, and sequence-to-sequence based model, respectively. Since BERT has fixed-length output limitation, we only investigate SA, PI, and NLI categories of tasks on it. Differently, BLOOM and T5 models have no fixed-length output limitation; thus, we investigate all tasks on them.

**Training Details of PET Methods**  We select four representative PET methods: Prompt (Lester et al., 2021), BitFit (Ben Zaken et al., 2022), Adapter (Houlsby et al., 2019a), and LoRA (Hu et al., 2022a), for conducting analysis experiments. To ensure the consistency of the PET methods' performance, we maintain the original design of each method, including the positions of tunable parameters and the number of trainable parameters, as reported in the respective original papers. Additionally, we train each PET method on 11 tasks using 3 different random seeds and report their average performance. Further details regarding the training configurations can be found in appendix B.

## 4.2 Model Scaling Impact on PET Methods

To investigate the impact of model scaling on PET methods, we arrange the Pre-trained Language Models (PLMs) in ascending order based on their model scale, and we report the performance of PET methods on each type of PLM.

Results are reported in Figure 2. First, we can observe that the PET methods exhibit noticeable performance differences (standard deviation

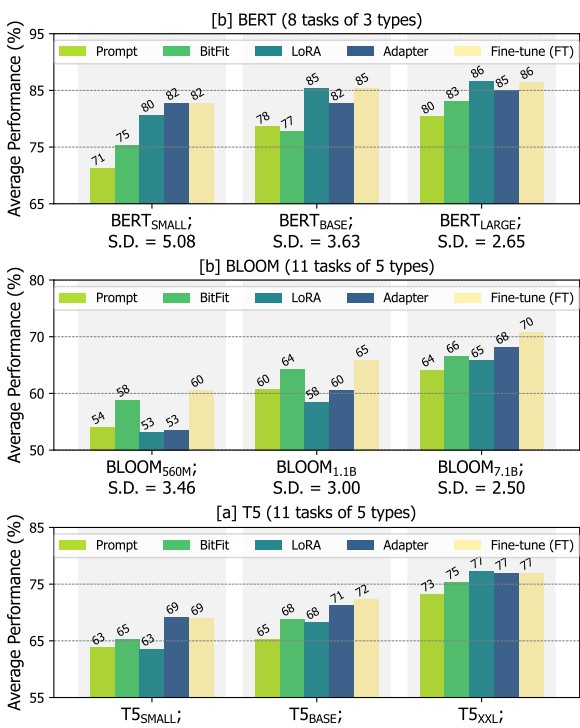

Figure 2: We investigate the average performance of the tuning methods, including Prompt, BitFit, LoRA, Adapter, and full-parameter fine-tuning, on three series of models. As the model scaling increases, the performance differences (standard deviation (S.D.)) among tuning methods become smaller.

(S.D.)) from each other on the general-scale models (BERT$_{SMALL}$ and BERT$_{BASE}$ in the sub-figure [a]; BLOOM$_{560M}$ and BLOOM$_{1.1B}$ in the sub-figure [b]; T5$_{SMALL}$ and T5$_{BASE}$ in the sub-figure [c]). This phenomenon is intuitive and demonstrates the critical impact of design differences (the position and quantity of parameters in the tunable module) on the performance of PET methods. This finding has been consistently found in numerous prior works (Ding et al., 2023; Hu et al., 2022c).

However, we find that as the model scaling increases (from BERT$_{SMALL}$ to BERT$_{LARGE}$ in the sub-figure [a]; from BLOOM$_{560M}$ to BLOOM$_{7.1B}$ in the sub-figure [b]; from T5$_{SMALL}$ to T5$_{XXL}$ in the sub-figure [c]), the performance discrepancies among PET methods diminish across all types of models, as evidenced by the decreasing standard deviation (S.D.) (from 5.08 to 2.65 on [a] BERT; from 3.46 to 2.50 on [b] BLOOM; from 2.75 to 1.72 on [c] T5). This finding implies that *the larger model scaling can mitigate the impact of the design differences among the PET methods on performance*.

## 5 Ablation Analyses

The design differences among the PET methods mainly lie in the tunable module's parameter position and parameter quantity. To further verify whether the model scaling will respectively remove the effects of the above differences on PET methods, we conducted two ablations to investigate whether model scaling can mitigate (1) the impact of tunable parameter position and (2) the impact of tunable parameter quantity.

However, only investigating the above four respective PET methods is insufficient to cover enough variations of parameter position for ablation study (1). This limitation makes us hard to preciously control the number of tunable parameters at the fine-grained (parameter level) in ablation study (2). Hence, we develop a more flexible PET method, **A**rbitrary **P**arameter-**E**fficient **T**uning (APET) method. Its tunable module can be arbitrary structure (§ 5.1) that facilitates us to explore various parameter positions in the ablation study (§ 5.2) and easier control the number of tunable parameters in the ablation study (§ 5.3).

### 5.1 Arbitrarily Parameter-Efficient Tuning (APET)

Similar to PET methods, the APET method is equipped with arbitrary module $\theta$ which is composed of $L$ tunable weights $\mathbf{W}$ distributed in any position of a model. Here, APET have three operations to insert the tunable weight $\mathbf{W}$ into any position of the PLM, thereby modify the specific layers and their corresponding transformations as follows:

**ADD** The tunable weight $\mathbf{W}$ will be into the PLM layer. The corresponding transformation of a PLM layer can be denoted as:

$$\mathbf{h}^{out} = f(\mathbf{h}^{in}) + \mathbf{W}_1. \qquad (4)$$

**CONCAT** The tunable weight $\mathbf{W}$ will be concatenated with the hidden state or the layer in the PLM. The corresponding transformation of a PLM layer can be denoted as:

$$\mathbf{h}^{out} = f(\mathbf{h}^{in}) + \begin{cases} f(\mathbf{W}_2) \\ \alpha \mathbf{h}^{in} \mathbf{W}_3 \mathbf{W}_4 \end{cases} \qquad (5)$$

**PLUG** The tunable weight $\mathbf{W}$ will be plugged between PLM layers. The corresponding transformation of a PLM layer can be denoted as:

$$\mathbf{h}^{out} = f(\mathbf{h}^{in}) + \sigma(f(\mathbf{h}^{in})\mathbf{W}_5)\mathbf{W}_6. \qquad (6)$$

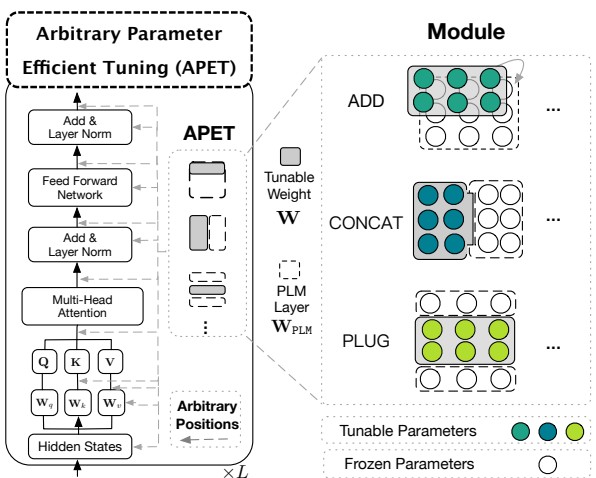

Figure 3: The tunable modules of APET methods ($\theta = \{\mathbf{W}_1, \mathbf{W}_2, ..., \mathbf{W}_L\}$) are composed $L$ tunable weights $\mathbf{W}$ with arbitrary structures. There are three operations (ADD, CONCAT, PLUG) for inserting these tunable weights $\mathbf{W}$ into a PLM.

Note that the inserted tunable weights $\mathbf{W}$ are not limited to the aforementioned structure as shown in Figure 3; they can be arbitrary structures. According to the inserted tunable weights and the corresponding modifications, the transformations of a PLM layer for APET method can be expressed as:

$$\mathbf{h}^{out} = f(\mathbf{h}^{in}) + \begin{cases} \mathbf{W}_1 \\ f(\mathbf{W}_2) \\ \alpha \mathbf{h}^{in} \mathbf{W}_3 \mathbf{W}_4 \\ \sigma(f(\mathbf{h}^{in})\mathbf{W}_5)\mathbf{W}_6 \\ \vdots \end{cases} \qquad (7)$$

By comparing Equation 7 with the equations of previously introduced Equation 2, it is obvious that the PET methods are special cases of APET method.

The module $\theta$ of APET are composed of arbitrarily inserted weights $\mathbf{W}$, which can be expressed as $\theta = \{\widehat{\mathbf{W}}_1, \widehat{\mathbf{W}}_2, ..., \widehat{\mathbf{W}}_L\}$. In the training process, we follow Equation (3) only to optimize $\theta$ while freezing the rest of the parameters ($\Phi$) in a PLM.

### 5.2 The Impact of Differences in Parameter Position on Performance

To investigate whether model scaling can mitigate the impact of parameter position in PET, we initially freeze other significant factors, i.e., the number of tunable parameters, that could potentially affect the performance. Given that the tunable parameters of the four aforementioned PET methods

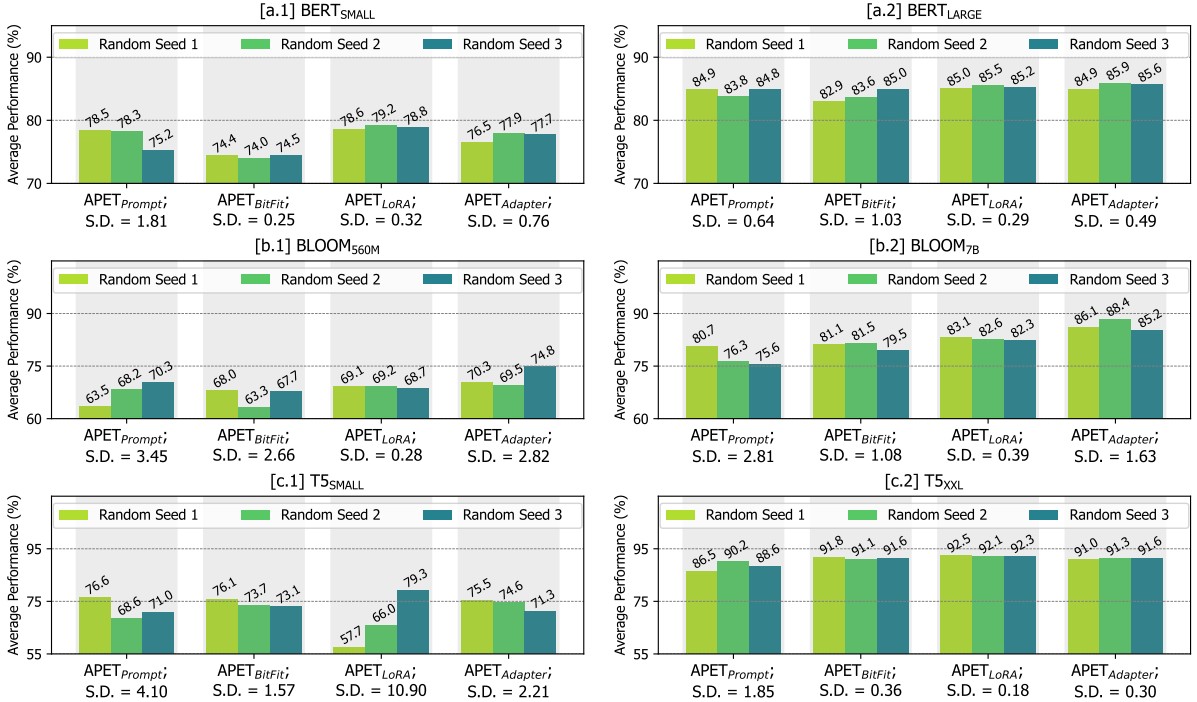

Figure 4: The parameter quantity in each group (bar: ) corresponds to the aforementioned four PET methods'. We denote the APET methods with the coresponding numbers of parameters as APET_Prompt, APET_BitFit, APET_LoRA, and APET_Adapter, respectively. Each APET method will arbitrarily select tunable parameters with different random seeds, each random seed representing a different parameter distribution. Here, S.D. means the standard deviation. As the model scaling increases, the impact caused by the parameter position on the performance becomes minor.

are fixed in the same positions, it is challenging for us to precisely conduct an experiment to assess the impact of position. Under this limitation, we then employ the APET method to arbitrarily select tunable parameters with different random seeds, each random seed representing a different parameter distribution, and train them on the tasks.

In the experiments, we set the number of tunable parameters for the APET methods in four groups. The parameter quantity in each group (bar: ) corresponds to that of the aforementioned four PET methods' (Prompt, BitFit, LoRA, Adapter). We denote these APET methods with varying numbers of parameters[3] as APET_Prompt, APET_BitFit, APET_LoRA, and APET_Adapter, respectively. Besides, we conduct the ablation study on three series of models (BERT, BLOOM, and T5) and report task (SST, RTE, and MRPC) average performance.

**Performance Comparison** As shown in Figure 4, there are four groups of comparisons in each sub-graph. We can observe that as a PLM size scales (BERT: from [a.1] to [a.2]; BLOOM:

from [b.1] to [b.2]; T5: from [c.1] to [c.2]), the performance differences (standard deviation (S.D)) of APET methods **within each group** decrease. Based on this findings, we argue that *larger models demonstrate greater effectiveness in mitigating the impact of differences in parameter position on performance*.

In addition, we have observed that despite the different number of tunable parameters **in four different groups** (bar: ) of APET methods, they have fewer performance differences on the larger model. We will delve into this finding further and provide an explanation for this phenomenon in § 5.3.

### 5.3 The Impact of Differences in The Number of Tunable Parameters on Performance

In this section, given the APET method under different numbers of tunable parameters, we observe their performance to conduct an ablation study.

From the reported results in Figure 5, we can find that (1) on the smaller models, e.g., BERT_SMALL (- -), BLOOM_560M (- - -), T5_SMALL (- - -) when the tunable parameters of tuning methods are fewer than a certain number, the performance will drop to randomly guess performance; (2) similarly,

---

[3]The number of tunable parameters are left in appendix D.

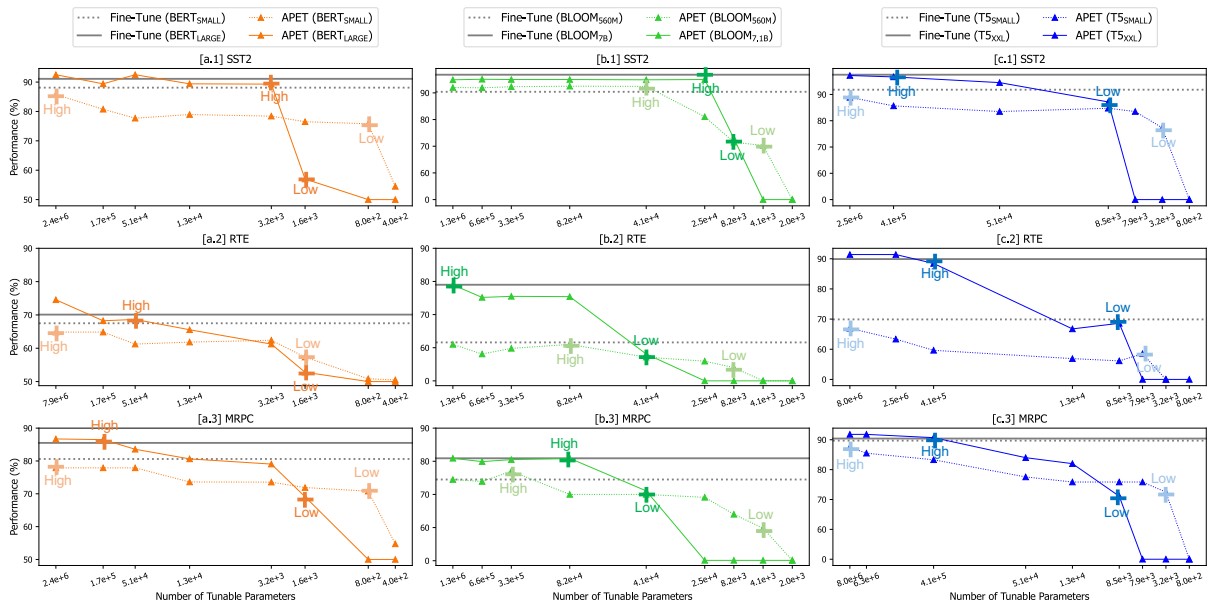

Figure 5: Given the different numbers of tunable parameters, we observe APET performance on three series of models and tasks. We find that (1) the model scaling can make tuning methods optimize fewer necessarily tuned parameters to reach full-parameter fine-tuning performance (- - - and ——); (2) APET methods require the similiar number of tunable parameters (low parameter thresholds lie the similiar range) to exceed random guess performance on the same models.

this phenomenon still holds on the larger models, BERT$_{\text{LARGE}}$ (——), BLOOM$_{7.1\text{B}}$ (——), T5$_{\text{XXL}}$ (—— ). Based on these findings, we can argue that that *model scaling cannot adequately eliminate the impact of the number of tunable parameters on the performance of PET methods.*

Interestingly, we find two parameter thresholds for tunable parameters in all models and name them as low parameter threshold (Low, Low, Low, Low, Low, Low) for necessary tuned parameters and the high parameter threshold (High, High, High, High, High, High) for necessary tuned parameters, respectively in Figure 5. When tunable parameters are more than *low parameter threshold,* the APET method can *exceed random performance* (e.g., $\frac{1 \times 100}{\text{Number of label types}}\%$ on BERT, 0% on BLOOM, and 0% on T5); when the tunable parameters are more than high parameter threshold, the APET method can almost achieve the full-parameter fine-tuning (FT) performance. Furthermore, we find that the model scaling affects the two parameter thresholds. Hence, we explore this phenomenon in the following paragraphs.

**High Threshold of Necessary Tuned Parameters** Based on the experimental results in the sub-graph [c.1] (SST2) of Figure 5, we find that the high threshold of the larger model is consistently lower

than the high threshold of the smaller model. This phenomenon holds true across all tasks (SST2, RTE, MRPC), and for all series of models, as depicted in all sub-graphs. Therefore, we can conclude that *model scaling enables tuning methods to train fewer necessary parameters while achieving the similar performance of full-parameter fine-tuning.*

This conclusion can intuitively explain why APET methods can achieve relatively similar performance on larger models, especially on T5$_{\text{XXL}}$, as illustrated in the aforementioned [c.2] in Figure 2. This is due to the fact that the number of tunable parameters in each group of APET methods surpasses the high parameter thresholds on T5$_{\text{XXL}}$; hence, they all achieve the similar performance of full-parameter fine-tuning.

**Low Threshold of Necessary Tuned Parameters** From the above results, we find that APET methods will exceed the random guess performance (0% on T5; 0% on BLOOM; 50% on BERT) and immediately reach the 80~90% full-parameter fine-tuning performance when the tunable parameters are more than low thresholds. However, the low thresholds are relatively higher on larger models (BERT$_{\text{LARGE}}$, BLOOM$_{7.1\text{B}}$, T5$_{\text{XXL}}$). Namely, APET methods require more tunable parameters to exceed the random guess performance. This phenomenon is con-

sistent over all tasks on all series of models. Hence, we can infer that the model scaling cannot reduce the number of necessary tuned parameters to drive PLMs to perform downstream tasks.

Furthermore, it is worth noting that *the low parameter thresholds of the APET methods almost lie in the same range on the same models*. Specifically, the range of low thresholds are in [8.0e+2, 3.2e+3] on $\text{BERT}_{\text{LARGE}}$, and [4.0e+2, 1.6e+3] on $\text{BERT}_{\text{SMALL}}$; [8.2e+3, 4.1e+4] on $\text{BLOOM}_{7.1B}$, [8.2e+3, 4.1e+3] on $\text{BLOOM}_{560M}$; [7.9e+3, 8.5e+3] on $\text{T5}_{\text{XXL}}$, [8.0e+2, 7.9e+3] on $\text{T5}_{\text{SMALL}}$. We will explain this phenomenon from the optimization perspective in § 6.

## 6 Discussing the Ablation Results from the Optimization Perspectives

The objectives of all parameter-efficient tuning methods (PET, APET) can be expressed as $\min_\theta \mathcal{L}(\mathcal{M}_{(\Phi,\theta)}(X), Y)$ as introduced in Equation (3), where $\theta$ is a tunable module. The module $\theta$ of different PET methods consists of different structures and varying numbers of tunable parameters. In this paper, we investigate the impact of model scaling on different modules, which possess varying numbers of tunable parameters distributed across multiple positions. We find that the larger model scaling can (1) mitigate the effects caused by the difference positions of tunable parameters (§ 5.2) and (2) make PET methods optimize fewer tunable parameters to achieve full-parameter fine-tuning performance (§ 5.3). To further fathom these phenomena, we will investigate the underlying reasons from an optimization perspective. (3) Besides, we also observe that PET methods can optimize almost the similar number of necessarily tuned parameters to exceed random guess performance on the same backbone models (§ 5.3). Although phenomenon (3) is not caused by model scaling, we can also explain it from the optimization perspective. Next, we together discuss it and the above two findings (1) and (2) in the following paragraphs.

**Why model scaling mitigates the effects caused by the differences in positions of tunable parameters on the PET performance?** From the optimal control perspective, a tunable module ($\theta$) of a tuning method can be seen as a controller (Yang and Liu, 2022; Ding et al., 2023) to drive PLMs towards downstream tasks. As the model scale increases, the larger model has higher parameter redundancy (Aghajanyan et al., 2021), allowing

arbitrary selection of tunable parameters for tuning without greatly degrading performance (Desai et al., 2019; Chen et al., 2020; Prasanna et al., 2020; Evci et al., 2020); thus, controllers (modules) might have higher degrees of freedom.

This might explain why the aribitray positions of the tunable parameters have less impact such that all PET methods can achieve the similar performance on the larger models. It is worth noting that even though the distribution of tunable parameters have less impact on the performance, it still affects converge speeds. Thus, finding a better parameter distribution to improve the converge speeds for PET methods is a direction worthy of exploring.

**Why model scaling leverages the fewer tunable parameters to achieve full-parameter fine-tuning performance?** Tuning $\theta$ to steer a PLM towards downstream NLP tasks can be seen as adaptations. From the perspective of representation space, the adaptations of PET methods can be re-parameterized into a unified low dimensional subspace (Qin et al., 2021; Aghajanyan et al., 2021; Yi et al., 2022). Aghajanyan et al. (2021) further demonstrate that adaptation on a larger PLM can be re-parameterized into the lower dimensional space; this implicitly explains why PET methods can optimize fewer parameters on larger-scale models, e.g., $\text{T5}_{\text{XXL}}$, to meet the full-parameter fine-tuning performance on tasks.

**Why can PET methods optimize the similar numbers of tunable parameters to exceed random guessing?** As stated above, the adaptations of the PET methods can be re-parameterized into a unified subspace. Qin et al. (2021) show that this low dimensional subspace is shared among all NLP tasks for the same PET methods. Yi et al. (2022) further suggest that this subspace is also shared among various PET methods. This might implicitly explain why all PET methods can tune the similar numbers of necessary tuned parameters to exceed the random guessing performance on the same models, even for the different tasks (§ 5.3).

## 7 Conclusion

The realm of model scaling for LLMs presents important and intriguing directions for the LLM community. The increasing of model scale unveils numerous emerging capabilities and advantages. In this work, our primary emphasis is on the impact of model scaling as it pertains to PET methods.

Through our comprehensive observation studies and in-depth discussions from optimization perspectives, we gain deeper insights into the effects of model scaling on PET and the reasons behind the observed phenomena. We believe that our findings will serve as a catalyst, inspiring further meticulous research and exploration in this area.

## 8 Limitations

This paper might have some possible limitations as follows: (1) we only explore the effects of the scaling law on performance. There might be other research points worth exploring, such as the power of model scale to convergence speed; (2) we study the power of model scale with comprehensive empirical experiments and explain the findings from the optimization perspective. There might be more theoretical proofs to explain these exciting findings.

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

## A  Task and Dataset

We use various NLP tasks to evaluate the APET methods, which can be divided into the following 5 categories:

**Sentiment Analysis (SA)**  SA tasks evaluate if a model can correctly predict the sentiment labels of an input sentence. In this paper, we choose SST-2 (Socher et al., 2013), IMDB (Maas et al., 2011), and Rotten Tomatoes (Pang and Lee, 2005).

**Natural Language Inference (NLI)**  NLI tasks evaluate a model's ability to correctly classify if a hypothesis can be entailed or not given a premise. In this paper, we choose MNLI (Williams et al., 2018), QNLI (Wang et al., 2019), and RTE (Bos and Markert, 2005).

**Paraphrase Identification (PI)**  PI tasks evaluate if a model can correctly identify paraphrases, which means two sentences are identical in semantic meaning. In this paper, we choose MRPC (Dolan and Brockett, 2005), and QQP (Sharma et al., 2019).

**Question Answering (QA)**  QA tasks evaluate a model's ability to answer questions. Context may be present. In this paper, we choose NQ-Open (Lee et al., 2019), an open-world QA dataset without context.

**Summarization (SUM)**  SUM tasks evaluate a model's ability to summarize a long paragraph into a shorter abstract without loosing the semantics of the original text. In this paper, we choose SamSUM (Gliwa et al., 2019), and Multi-News (Fabbri et al., 2019) in our experiments.

## B  Parameter-efficient Tuning (PET) Methods

Here, we first recap the PLM (transformer) layer. Then, we describe the detail and training configurations of the PET methods shown in Figure 1.

### B.1  Transformer Architecture

A PLM is generally a stack of multiple Transformer layers, each composed of a multi-headed attention and a feed-forward network. The multi-headed attention contains $h$ attention heads working in parallel. Specifically, given an input $\mathbf{X} \in \mathbb{R}^{n \times d}$, the $i$-th attention head works as follows:

$$\mathbf{h}_i = \mathsf{softmax}(\frac{(\mathbf{X}\mathbf{W}_q^i)(\mathbf{X}\mathbf{W}_k^i)^T}{\sqrt{d/h}}(\mathbf{X}\mathbf{W}_v^i)), \quad (8)$$

where $n$ is sequence length, $d$ is the hidden dimension, $\mathbf{W}_q^i \in \mathbb{R}^{n \times d}$ is query, $\mathbf{W}_k^i \in \mathbb{R}^{n \times d}$ is key, and $\mathbf{W}_v^i \in \mathbb{R}^{n \times d}$ is value. The output from each attention head will be concatenated and further transformed by $\mathbf{W}_o \in \mathbb{R}^{d \times d}$ and be denoted as:

$$\mathbf{h}_{\mathsf{MHA}} = \mathsf{concat}(\mathbf{h}_1, \mathbf{h}_2, ..., \mathbf{h}_h)\mathbf{W}_o, \quad (9)$$

where $\mathbf{h}_{\mathsf{MHA}} \in \mathbb{R}^{n \times d}$ is the output hidden state of multi-headed attention layer. After that, $\mathbf{h}$ will be fed into a two-layer feed-forward network

$$\mathbf{h}_{\mathsf{FFN}} = \sigma(\mathbf{h}\mathbf{W}_1 + \mathbf{b}_1)\mathbf{W}_2 + \mathbf{b}_2, \quad (10)$$

where $\mathbf{W}_1 \in \mathbb{R}^{d \times d_m}$, $\mathbf{W}_2 \in \mathbb{R}^{d_m \times d}$, $\mathbf{b}_1 \in \mathbb{R}^{d_m}$, $\mathbf{b}_2 \in \mathbb{R}^d$, and $d_m > d$ is an integer.

During the forward pass through each (transformer) block, the input hidden state is applied with the sequence of layers. For simplicity, we formalize the transformation of each layer as

$$\mathbf{h}^{out} = f(\mathbf{h}^{in}). \quad (11)$$

Under the layer as the operator $f$, the input hidden state $\mathbf{h}^{in} \in \mathbb{R}^{n \times d}$ is transformed into the output hidden state $\mathbf{h}^{out} \in \mathbb{R}^{n \times d}$, where $s$ is the input length, and $d$ is the dimension.

### B.2  Implementation Details of PET Methods

**Prompt**  Prompt-tuning (Lester et al., 2021) prepends $N_p$ tunable soft tokens, i.e. embeddings, to the input sentences and asks the model to predict the probability of the next word. During training, only the newly added embeddings are optimized and the backbone model is frozen.

**BitFit**  BitFit (Ben Zaken et al., 2022) is a method that only tunes all the bias terms $\mathbf{W}_b \in \mathbb{R}^d$ in the PLM, which lie in the self-attention and layer norm layers.

**LoRA**  LoRA (Hu et al., 2022b) is a method that adapts a PLM in a low-rank space. It down-projects the attention weights into a lower dimension and up-projects it back to the original dimension. Only these projection weights are optimized.

**Adapter**  Adapter (Houlsby et al., 2019b) is a method that only tunes the inserted adapter modules, which consist of down projection, non-linear transformation, up projection, and a skip-connection. For each existing Transformer layer in a PLM, the adapter modules are inserted at two

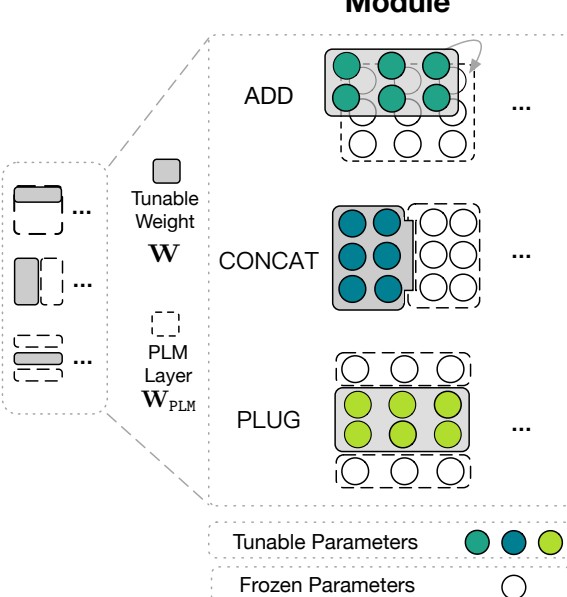

**Module**

ADD

Tunable
Weight
$\mathbf{W}$

CONCAT

PLM
Layer
$\mathbf{W}_{\text{PLM}}$

PLUG

Tunable Parameters

Frozen Parameters

Figure 6: The tunable modules of APET methods are composed $p$ tunable weights $\mathbf{W}$, which can be expressed as $\theta = \{\mathbf{W}_1, \mathbf{W}_2, ..., \mathbf{W}_L\}$. We introduce three operations to insert the tunable weight $\mathbf{W}$ into the PLM and the corresponding transformation.

locations: (1) after the first feed-forward layer, and (2) after the two consecutive feed-forward layers. During training, only the adapter modules are optimized and the rest of the PLM is frozen.

### B.3  Training Configurations of PET Methods

The tunable module of a PET method $\theta$ is composed of $L$ tunable weights $\mathbf{W}$ (all tunable weights) of the specific PET method, which can be expressed as $\theta = \{\mathbf{W}_1, \mathbf{W}_2, ..., \mathbf{W}_L\}$. We also follow Equation (3) to train the PET method. During training, we only optimize $\theta$ while freezing the rest of the parameters in the PLM. We adopt a batch size of 32 and have no warm-up for most of the PET models and tasks. The maximum input length is 128 for single sentence tasks (SA) and 256 for multi-sentence tasks (NLI, PI, QA, SUM). The maximum generation length is 1 for classification tasks (SA, NLI, PI), 64 for Multi-News, and 128 for SAM-Sum. On the BERT, BLOOM, T5 models, we set their learning rates as {3e-4}, {3e-4, 5e-5}, {1e-4, 1e-3, 1e-2} respectively. Then, we choose the best performance to report.

## C  Arbitrary Parameter-Efficient Tuning (APET) Methods

We introduce a more flexible PET method, **A**rbitrary **P**arameter-**E**fficient **T**uning (APET) method. Its tunable module can be arbitrary structure that facilitates us to explore various module structures (parameter position) and easier control the number of tunable parameters.

### C.1  Implementation Details of APET Methods

As we previously introduced in § 5.1, the tunable module of the APET method is composed of tunable weights. Each tunable weight can be expressed as $\mathbf{W}$. Here, we have three operations to insert the tunable weight $\mathbf{W}$ into the PLM to modify the specific layers and their corresponding transformations as follows:

**ADD**  We will add the tunable weight $\mathbf{W}$ into the PLM layer. The corresponding transformation can be denoted as $\mathbf{h}^{out}$:

$$f(\mathbf{h}^{in}) + \mathbf{W}_1. \tag{12}$$

**CONCAT**  We will concatenate the tunable weight $\mathbf{W}$ and the hidden state or the layer in the PLM. The corresponding transformation can be denoted as $\mathbf{h}^{out}$:

$$f(\mathbf{h}^{in}) + \begin{cases} f(\mathbf{W}_2) \\ \alpha \mathbf{h}^{in}\mathbf{W}_3\mathbf{W}_4 \end{cases} \tag{13}$$

**PLUG**  We will plug the tunable weight $\mathbf{W}$ between PLM layers. The corresponding transformation can be denoted as $\mathbf{h}^{out}$:

$$f(\mathbf{h}^{in}) + \sigma(f(\mathbf{h}^{in})\mathbf{W}_5\mathbf{W}_6). \tag{14}$$

According to these operations and the corresponding transformations, we can express the APET methods as $\mathbf{h}^{out}$:

$$f(\mathbf{h}^{in}) + \begin{cases} \mathbf{W}_1 \\ f(\mathbf{W}_2) \\ \alpha \mathbf{h}^{in}\mathbf{W}_3\mathbf{W}_4 \\ \sigma(f(\mathbf{h}^{in})\mathbf{W}_5)\mathbf{W}_6 \\ \vdots \end{cases} \tag{15}$$

By comparing the Equation (15) with the equations of the previously introduced PET methods, we can clearly find that the PET methods are special cases of APET methods.

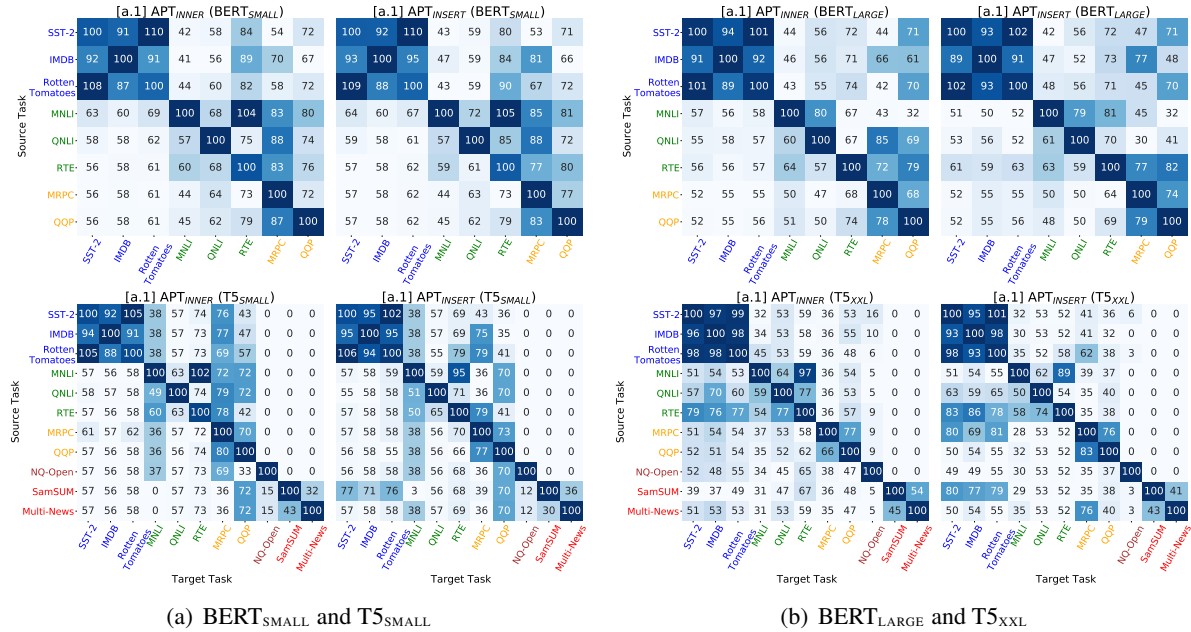

|  | (a) BERT$_{SMALL}$ and T5$_{SMALL}$ | (b) BERT$_{LARGE}$ and T5$_{XXL}$ |

Figure 7: Relative performance (zero-shot transfer performance / original performance) (%) on the target tasks (columns) of the APET methods trained on the source tasks (rows). Colors of the task names indicate task types.

## C.2 Training Configurations of APET methods

The tunable module of a APET method $\theta$ is composed of $L$ tunable weights $\mathbf{W}$, which can be expressed as $\theta = \{\mathbf{W}_1, \mathbf{W}_2, ..., \mathbf{W}_L\}$. We also follow Equation (3) to train the APET method. During training, we only optimize $\theta$ while freezing the rest of the parameters in the PLM.

Besides, we adopt a batch size of 32 and have no warm-up for most of the APET models and tasks. In addition, The maximum input length is 128 for single sentence tasks (SA) and 256 for multi-sentence tasks (NLI, PI, QA, SUM). The maximum generation length is 1 for classification tasks (SA, NLI, PI), 64 for Multi-News, and 128 for SAMSum. On the BERT, BLOOM, T5 models, we set their learning rates as {3e-4}, {3e-4, 5e-5}, {1e-4, 1e-3, 1e-2} respectively. Then, we choose the best performance to report.

## D Number of Tunable Parameters of APET

Here, the Table 2 shows the number of tunable parameters of APET for each group in Figure 4.

## E Power of Model Scale to Transferability

Furthermore, to explore whether the power of model scale can also facilitate generalization ability of tuning methods, we explore the transferability

|  | APET$_{Prompt}$ | APET$_{BitFit}$ | APET$_{LoRA}$ | APET$_{Adapter}$ |
|---|---|---|---|---|
| BERT$_{SMALL}$ | 5.1e+4 | 1.4e+4 | 6.6e+4 | 2.0e+5 |
| BERT$_{LARGE}$ | 1.0e+5 | 1.7e+5 | 7.9e+5 | 2.4e+6 |
| BLOOM$_{560M}$ | 1.0e+5 | 2.7e+5 | 7.9e+5 | 2.4e+6 |
| BLOOM$_{7.1B}$ | 4.1e+5 | 1.4e+6 | 3.9e+6 | 1.2e+7 |
| T5$_{SMALL}$ | 5.1e+4 | 1.2e+5 | 2.3e+5 | 8.0e+5 |
| T5$_{XXL}$ | 4.1e+5 | 2.5e+6 | 6.3e+6 | 1.9e+7 |

Table 2: The table shows the numbers of tunable parameters of tunable parameters in Figure 4 experiment.

between the NLP tasks in the zero-shot setting (Vu et al., 2022; Su et al., 2022; Ding et al., 2023). In the experiments, we first train the parameters of APET methods on the source tasks and directly reuse them on the target tasks in zero-shot setting. We will investigate two series of PLMs T5 (T5$_{SMALL}$ and T5$_{XXL}$) and BERT (BERT$_{SMALL}$ and BERT$_{LARGE}$) and report the relative performance.

Note that for different types of tasks, they are expected to share different groups of label sets (e.g. for task like SA, the labels are usually positive/negative, whereas, for tasks like NLI, the labels are usually entailment/not entailment). Reusing the parameters trained on the source task to test on the target task will naturally fail since the model is not able to generate the labels they have never seen in the training stage. To this end, we generally map the original label sets to a unified label

set (e.g. negative/not entailment/false –> 0, positive/entailment/true –> 1). Utilizing a unified label set makes it feasible to evaluate the transferability of the AFP method among different types of tasks regardless of the divergence of original labels.

The results are shown in Figure 7, from which we can find that the APET (APET$_{\text{DISCRETE}}$ and APET$_{\text{ADJACENT}}$) methods can transfer to the same type of tasks demonstrated by the darker color alongside the diagonal of the matrix and generally perform well both on small-scale PLMs (Figure 7 (a): BERT$_{\text{SMALL}}$ and T5$_{\text{SMALL}}$) and large-scale PLMs (Figure 7 (b): BERT$_{\text{LARGE}}$ and T5$_{\text{XXL}}$). However, the lighter color indicates that APET methods have difficulty performing different types of tasks overall, and both small-scale and large-scale PLMs share this phenomenon. This finding indicates that the power of scale does not necessarily facilitate the generalization ability of AFP methods which is in line with the prevalent assumption that fewer parameters often cause underfitting, whereas more parameters tend to cause overfitting. Nevertheless, the mechanism behind this phenomenon still arouses our deep concern and is worth expanding that we will systematically analyze it in our future work.