# OpenReview forum: "Exploring the Impact of Model Scaling on Parameter-Efficient Tuning"
_EMNLP/2023/Conference — EMNLP 2023 Main_

### Official Review · Reviewer_rQAh · 2023-07-24

**Soundness:** 3

**Excitement:**

2: Mediocre: This paper makes marginal contributions (vs non-contemporaneous work), so I would rather not see it in the conference.

**Paper Topic And Main Contributions:**

This paper focuses on parameter-efficient tuning. Specifically, they firstly study the impact of model scaling on different parameter-efficient tuning (PET) methods and then propose the arbitrarily PET to investigate the impact of tunable parameters positions and quantity.

**Questions For The Authors:**

1. In Table 1, for prompt method, is it right that $\Delta h=f(W_{prompt})$? If it is right, then for every input (also can be represented by $h^{in}$), the $\Delta h=f(W_{prompt})$ is always the same.  I think $f$ should be a function of both $W_{prompt}$ and input.

**Reasons To Accept:**

1.  The paper is well written and well structured.
2. The author concludes the relationship bewteen tunable parameter positions, quantity and model scaling, although they are a bit obvious.

**Reasons To Reject:**

1. It seems that the APET they proposed is just a combination of different PET methods, which is also introduced in the paper "Towards a Unified View of Parameter-Efficient Transfer Learning". Therefore, it should not be regarded as a contribution.
2. Some parts of the paper are not described clearly. For example, in Section 5.2, to investigate the position impact,  the number of tunable parameters should be fixed. How do you achieve this with APET? Can you describe it more detailly? Moreover, most of the analysis focuses on the "within group comparision. I think this section should focus on comparisions between different groups.

**Reproducibility:**

4: Could mostly reproduce the results, but there may be some variation because of sample variance or minor variations in their interpretation of the protocol or method.

**Reviewer Confidence:**

4: Quite sure. I tried to check the important points carefully. It's unlikely, though conceivable, that I missed something that should affect my ratings.

---

> ### Author Rebuttal · Authors · 2023-08-27
>
> Dear Reviewer, I sincerely appreciate your insightful comments.  After reviewing the main contribution that you outlined, I have taken note that certain key information may have been inadvertently misunderstood or omitted. Consequently, the Area Chair (AC) has kindly informed me and suggested that I engage in a prior discussion with you through official comments.
>
> Response to Reasons to reject:
> - Response [1]:
> Firstly, I would like to emphasize the primary objective of our work, as clearly highlighted in our title: "Exploring the Impact of Model Scaling on Parameter-efficient Tuning". Our primary contribution revolves around the exploration of model scaling, not the introduction of APET as a novel method.
> Secondly, I would like to clarify a crucial aspect:
> Our intention with APET was never to present it as a mere combination of existing PET methods. Instead, APET is utilized as an analytical tool in our work, aiding us in delineating our main observations.
> The core contribution of our paper lies in the observations we derived using APET:
>
>     - 1.The influence of tunable parameter positions on performance diminishes with **model scaling**.
>
>     - 2.**Larger** models can achieve performance comparable to full-parameter fine-tuning by optimizing a reduced number of tunable parameters, in contrast to smaller models.
>
> - Response [2]: The tunable module (comprising a varying number of tunable parameters) within APET can adopt arbitrary shapes and dimensions. Consequently, it becomes convenient to regulate the overall count of tunable parameters to. I am committed to incorporating this description into the paper. Additionally, for a more comprehensive understanding, I encourage you to refer to the appendix and the provided code, which offer further visualizations and detailed insights.
>
> I sincerely appreciate the valuable feedback provided by the reviewer. I am hopeful that these responses will aid in conveying the contributions of this paper more comprehensively and also help in addressing your concerns. Feel free to let me know if you have any further questions via official comments, I will reply you ASAP.

---

### Official Review · Reviewer_fvk4 · 2023-08-03

**Typos Grammar Style And Presentation Improvements:** Line 123
**Soundness:** 4

**Excitement:**

3: Ambivalent: It has merits (e.g., it reports state-of-the-art results, the idea is nice), but there are key weaknesses (e.g., it describes incremental work), and it can significantly benefit from another round of revision. However, I won't object to accepting it if my co-reviewers champion it.

**Missing References:**

1. Mao Y, Mathias L, Hou R, et al. UniPELT: A Unified Framework for Parameter-Efficient Language Model Tuning.

2. Zhou H, Wan X, Vulić I, Korhonen A. AutoPEFT: Automatic Configuration Search for Parameter-Efficient Fine-Tuning.

3. Hu S, Zhang Z, Ding N, et al. Sparse Structure Search for Parameter-Efficient Tuning.

4. Chen J, Zhang A, Shi X, Li M, Smola A, Yang D. PARAMETER-EFFICIENT FINE-TUNING DESIGN SPACES.

5. Lawton N, Kumar A, Thattai G, Galstyan A, Steeg GV. Neural Architecture Search for Parameter-Efficient Fine-tuning of Large Pre-trained Language Models.

6. Liu H, Tam D, Muqeeth M, et al. Few-Shot Parameter-Efficient Fine-Tuning is Better and Cheaper than In-Context Learning.

7. Giannou A, Rajput S, Papailiopoulos D. The Expressive Power of Tuning Only the Norm Layers.

**Paper Topic And Main Contributions:**

This paper provides a study of the performance of different PEFT methods across different scales of models. It describes interesting findings that model scaling can mitigate the effects of insertion positions of PEFT and the trainable parameter is a critical factor for achieving competitive performance for PEFT in comparison to FFT. It is delightful to read this paper, and the presentation, visuals, of this paper is great. However, the APET framework for studying the parameters is not new. Hence, the technical contribution of this paper is not the strength of this paper, but I appreciate the analysis of PEFT across scales for driving PEFT research further.  I will recommend the author to include the detailed performance of each experiment for each task in appendix for providing additional information rather than only providing an averaged number in the main section. The experiments are solid for the discussed 4 PEFT methods for deriving the corresponding findings and the results are as expected from PEFT researchers, but I expect for more PEFT modules to be studied. This work can give more technical contributions to the field if the author could also study the effectiveness of mixture of different insertion methods in a limited parameter budget, layer positions for inserting PEFTs, and the most recommended/universal PEFT solution for large language models. In conclusion, I am learning positive for this work given its experimental solidness, and will recommend for a findings-level of work given the limited technical contribution.

**Questions For The Authors:**

1. If the author could address the concerns in the weakness section, I will further improve my scores, especially for point 3, 4.
2. Prefix-tuning (prompt tuning) can leverage reparameterization, and the parameter amount learned by additional MLP layers can give significant impacts to the performance. Have you considered studying this?

---------------------
### After Rebuttal

I appreciate the response from the author. I am convinced now that the current experiments are solid for delievering the findings. I have increased my score for soundness and keep the excitement score unchanged. I appeciate if the author could include more PEFT results in the camera-ready version.

**Reasons To Accept:**

1. The experimental results are solid and as expected.
2. The writing and presentation of this work is great.
3. It gives useful empirical findings for PEFT researchers facing different scales of language models.

**Reasons To Reject:**

1. The technical contribution of this paper is limited.
2. The APET framework is related with other arbitrary PEFT frameworks: UniPELT [1], AutoPEFT [2], S3PET [3], PEFT Design Space [4], NASPEFT [5], but they are not cited nor discussed.
3. The technical part of this paper can be improved by studying the effectiveness of mixture of different insertion methods and the most recommended PEFT solution for large language models.
4. As an analysis paper, I will expect more types of PEFT modules to be studied in this paper, such as Compactor, IA3 [6], LayerNorm Tuning [7], Parallel Adapter. It will be interesting to see if the current conclusion applies to wider ranges of PEFT modules.

**Reproducibility:**

4: Could mostly reproduce the results, but there may be some variation because of sample variance or minor variations in their interpretation of the protocol or method.

**Reviewer Confidence:**

4: Quite sure. I tried to check the important points carefully. It's unlikely, though conceivable, that I missed something that should affect my ratings.

---

> ### Author Rebuttal · Authors · 2023-08-27
>
> Dear Reviewer,
> I sincerely appreciate your insightful comments.
>
> Response to Reasons to reject:
>
> - Response [2]: Actually, I have referenced certain papers in my work, such as UniPELT [1], S3PET [3], and so forth. However, some of citations correspond to earlier versions on Arxiv, which might lead to discrepancies. I am committed to rectifying this by updating the citations and incorporating any omitted references.
>
> - Response [3]: Yes, you are right. Actually, you can observe that we have also undertaken experiments concerning this aspect, depicted in Figure 4. Within this figure, APET is presented as a fusion of various insertion methods, with the adjustable parameters corresponding to each PET method expounded in the paper. Therefore, if you have an interest in exploring the efficacy of the amalgamation of different insertion methods, kindly refer to Figure 4. Doing so, you will lead you to the same findings: with the increasing of model scale, the performance variation among APET methods become smaller.
>
> - Response [4]: Actually, although I mentioned Prompt, Adapter , LoRA, and BitFit in the paper, I cover more various and arbitrary PET methods in our implementation with APET. Hence, the tunable modules, can various structures and be distributed in any positions. I believe the continuations of APET must include Compactor, IA3 [6], LayerNorm Tuning [7], Parallel Adapter, and more (you could further refer to our appendix and the code). Since, PET methods are continuously proposed by months, I am hard to list all of them in the paper. Hope this implementation detail can mitigate your concerns and could understand.
>
> Actually, although in the paper, I only mentioned Prompt, Adapter, LoRA, and BitFit as examples, it's worthy to note that the implementation of APET encompass a wider array of diverse and arbitrary PET methods (its the tunable modules, can various structures and be distributed in any positions). Consequently, its modules can assume various structures and be positioned flexibly throughout the architecture. I am confident that the configurations of APET encompass methods such as Compactor, IA3 [6], LayerNorm Tuning [7], Parallel Adapter, and more (for further details, you can refer to our appendix and the provided code).
>
> As the landscape of PET methods continues to evolve over time, add every single method within the implementation becomes a challenging endeavor. But I've already covered all of them as more as possible in implementation. I hope that this level of implementation detail addresses your concerns and aids in your understanding.
>
>
> Questions For The Authors:
> - Response[1]: Please refer to the aforementioned Response [3] and Response [4] in response to reasons to reject. If I do not reply well or you have further questions, please kindly let me know via official comments.
>
> - Response[2]: Yes, indeed, I have done so. In fact, I have also undertaken Prefix-tuning across three different series of LLMs and 9 distinct tasks. The outcomes can be presented in Figure in the paper as follows:
> BERT_{SMALL}: 73, BERT_{BASE}: 81, BERT_{LARGE}: 82
> BLOOM_{560M}: 55, BLOOM_{1.1B}: 59, BLOOM_{7.1B}: 65
> T5_{SMALL}: 64, T5_{BASE}: 63, T5_{XXL}: 76
> However, due to constraints in computational resources, I was able to conduct Prefix-tuning performance evaluations on T5-XLL across 9 tasks (the above T5_{XXL}: 76 are evaluated on only 9 tasks), as opposed to the intended 11 tasks. Regarding the aspect of thoroughness, while I have refrained from presenting the Prefix-tuning outcomes in the main body, I plan to further include these results within the appendix for your reference. Furthermore, concerning the implementation of APET, I would like to highlight that we integrate both prompt and Lora modules. This combination can inherently encompasses the Prefix-tuning methods as well. For a more comprehensive understanding, I encourage you to consult the provided code.
>
> I sincerely appreciate the valuable feedback provided by the reviewer. I am hopeful that these responses will aid in conveying the contributions of this paper more comprehensively and also help in addressing your concerns. Feel free to let me know if you have any further questions via official comments, I will reply you ASAP.

---

### Official Review · Reviewer_8CW5 · 2023-08-05

**Typos Grammar Style And Presentation Improvements:** Section 5.3 and in particular l484-49…
**Soundness:** 3

**Excitement:**

4: Strong: This paper deepens the understanding of some phenomenon or lowers the barriers to an existing research direction.

**Paper Topic And Main Contributions:**

The paper addresses an important topic, particularly parameter efficient tuning (PET) through the use of techniques that add a handful of parameters to (very) large models and optimize only those parameters on domain/task specific data sets in an incremental training stage.

Such techniques have been shown to be able to achieve the same performance gains as full model fine-tuning/incremental training on the data set of interest and so are very attractive particularly when it comes to serving large models: the base model parameters can be re-used across many tasks, and only the task adaptation "stub" needs to be swapped when serving requests of a known, specific type.

**Reasons To Accept:**

The authors present a thorough study of various PET techniques, and also propose a novel one to set the experimental framework on a solid footing.

**Reasons To Reject:**

I could not fully understand the case for invariance of low/high parameter thresholds, so I am not ready to accept the resulting conclusions.

**Reproducibility:**

4: Could mostly reproduce the results, but there may be some variation because of sample variance or minor variations in their interpretation of the protocol or method.

**Reviewer Confidence:**

4: Quite sure. I tried to check the important points carefully. It's unlikely, though conceivable, that I missed something that should affect my ratings.

---

> ### Author Rebuttal · Authors · 2023-08-27
>
> Dear Reviewer,
> I sincerely appreciate your insightful comments.
> After reviewing the main contribution that you outlined, I have taken note that certain key information may have been inadvertently misunderstood or omitted. Consequently, the Area Chair (AC) has kindly informed me and suggested that I engage in a prior discussion with you through official comments.
>
> I will illustrate the main contributions presented in this paper and provide a more clear explanation of Section 5.3 to address your concerns. Hope the following explanations can make you better understand Section 5.3 and better evaluate the contributions of this work.
>
>
> Contribution:
> The main contribution of this paper is to identify that model scaling (larger models) can (1) mitigate the effects of the positions of tunable parameters on performance, and (2) enables tuning methods to achieve performance comparable to full-parameter fine-tuning by optimizing fewer tunable parameters.
>
> Question Response:
> Regarding Section 5.3, this section demonstrates the phenomenon in the aforementioned (2). Specifically, we found that that all parameter-efficient tuning (PET) methods can optimize fewer tunable parameters to achieve the full-parameter fine-tuning performance (upper bound performance). Besides, we can find that various PET methods exhibit two distinct parameter thresholds: a low threshold and a high threshold. When the number of tunable parameters reaches the low threshold, the performance of the PET method surpasses that of random guessing. When reaching the high threshold, the PET method achieves performance comparable to full-parameter tuning. Interestingly, we observed that when different PET methods are applied to the same LLM backbone, their parameter low thresholds consistently lie within the range mentioned in Section 5.3 (lines 484-493).
>
> I trust that this information will assist you in comprehending this study and addressing any concerns you may have. Lastly, kindly hope that you consider re-evaluating this work. If you have any questions or problems further would like to discuss, please let me know. I will reply you as soon as possible here.

---

### Meta-Review · Area_Chair_daEi · 2023-09-19

**Recommendation:** 5

**Metareview:**

This work focuses on parameter-efficient tuning (PET) and the impact of the pretrained model size on PET methods and design differences. The authors found that as the pretrained model scale increases, the performance differences among PET methods decrease. They explore the impact of model scale in terms of the position and the number of tunable parameters.

The authors present a comprehensive study including several tasks and different PET methods/designs in the form of a unified framework (APET) for analysis. As reviewers commented, the paper presents empirical findings based on solid experimental results that are helpful for researchers who study PET methods. Furthermore, I believe authors will incorporate the reviewers suggestions and points together with their adiditonal results to the final version of the paper.

---

### Decision · Program_Chairs · 2023-10-07

**Decision:**

Accept-Main

**Comment:**

This work focuses on parameter-efficient tuning (PET) and the impact of the pretrained model size on PET methods and design differences. The authors found that as the pretrained model scale increases, the performance differences among PET methods decrease. They explore the impact of model scale in terms of the position and the number of tunable parameters.

The authors present a comprehensive study including several tasks and different PET methods/designs in the form of a unified framework (APET) for analysis. As reviewers commented, the paper presents empirical findings based on solid experimental results that are helpful for researchers who study PET methods. Furthermore, I believe authors will incorporate the reviewers suggestions and points together with their adiditonal results to the final version of the paper.